# Computation of Evapotranspiration with Artificial Intelligence for Precision Water Resource Management

**Hassan Afzaal [1] , Aitazaz A. Farooque [1],\*, Farhat Abbas [1],\* , Bishnu Acharya [1] and Travis Esau [2]**

[1] Faculty of Sustainable Design Engineering, University of Prince Edward Island, Charlottetown, PE C1A4P3, Canada; hafzaal2@upei.ca (H.A.); bacharya@upei.ca (B.A.)

[2] Engineering Department, Dalhousie University, Agriculture Campus, Truro, NS B2N5E3, Canada; tesau@dal.ca

\* Correspondence: afarooque@upei.ca (A.A.F.); fabbas@upei.ca (F.A.); Tel.: +1-902-566-6084 (A.A.F.)

**Abstract:** Accurate estimation of reference evapotranspiration (ETo) provides useful information for water resource management and sustainable agriculture. This study estimates ETo with recurrent neural networks (RNNs), namely long short-term memory (LSTM) and bidirectional LSTM. Four representative meteorological sites (North Cape, Summerside, Harrington, and Saint Peters) were selected across Prince Edward Island (PEI), Canada to form a PEI dataset from mean values of the four sites' climatic variables for capturing climatic variability from all parts of the province. Based on subset regression analysis, the highest contributing climatic variables, namely maximum air temperature and relative humidity, were selected as input variables for RNNs' training (2011–2015) and testing (2016–2017) runs. The results suggested that the LSTM and bidirectional LSTM are suitable methods to accurately ($R^2 > 0.90$) estimate ETo for all sites except Harrington. Testing period (2016–2017) root mean square errors were recorded in range of 0.38–0.58 mm/day for all sites. No major differences were observed in accuracy of LSTM and bidirectional LSTM. Another objective of this study was to highlight the potential gap between $ET_O$ and rainfall for assessing agriculture sustainability in Prince Edward Island. Analyses of the data highlighted that the cumulative ETo surpassed the cumulative rainfall potentially affecting yield of major crops in the island. Therefore, agriculture sustainability requires viable options such as supplemental irrigation to replenish the crop water requirements as and when needed.

**Keywords:** recurrent neural networks; deep learning; irrigation scheduling; Penman–Monteith; physical hydrology components; water cycle budgeting

## 1. Introduction

Evapotranspiration is a key element in water balance as well as in the surface energy equation. Accurate estimation of ETo provides useful information for water resource management, irrigation scheduling, and crop sustainability. Lysimeters are commonly used to directly estimate ETo; however, the use of lysimeters for ETo estimation is very limited because of high maintenance and operational costs on lysimeters [1]. Several mathematical models indirectly estimate ETo and are considered to be the intelligent alternative of direct methods due to time saving and ease of application [2]. Penman–Monteith (FAO-56) is the most acceptable mathematical model for estimation of ETo [3].

Under both humid and arid climatic conditions, the FAO-56 method has been unanimously reported to be the most efficient method for ETo estimation by incorporating thermodynamic as well as aerodynamic effects [4]. However, input data needed by the FAO-56 method including

temperature, relative humidity, solar radiation, wind speed, and more information about the area make its applicability challenging for several locations across the globe. Solutions to this problem have been sought by introducing various empirical methods through simplifying the FAO-56 method; such as, Hargreaves equation that requires temperature data only to estimate ETo. The choice of methods solely depends upon the accuracy of methods and availability of reliable data. An ideal method, however, should be based on minimal input data variables with no compromise on precision and accuracy [5].

Artificial neural networks (ANNs) have drawn the attention of researchers to model several complex non-linear hydrological relationships. Several ANNs have been successfully used to solve hydrology related problems, such as river flows extrapolation [6], rainfall run-off modeling [7], sediment forecasting [8], and notably ETo [9]. Afterwards, several improvements in ANN's architecture, learning algorithms have been proposed by different researchers. For example, Sudheer et al. [10] modeled the ETo for rice crop. They used radial basis neural networks with varied combinations of climatic input variables. Aytek et al. [11] proposed the explicit neural network to model ETo by using daily climatic variables in California, USA after comparing six different conventional methods. Rahimikhoob [12] trained the ETo models by using only the air temperature of the Caspian Sea in North Iran and compared the results with the FAO-56 model. He concluded that the air temperature was able to explain the variability in ETo without compromising the accuracy of results. In recent years, several other machine learning models were tested to estimate ETo such as extreme learning machines [2,13,14], support vector machines [10], and fuzzy genetic approach [15].

Several climatic variables such as temperature, relative humidity, and ETo exhibit seasonality and may be treated as a time series problem. Although, ANNs handle the non-linear behavior of time series better than regular regression; however, most of the ANNs do not explain seasonality and time dependence. Simple ANN architecture such as multilayer perceptron does not contain any memory blocks to store the previous information for better prediction. To address this issue, recurrent neural networks (RNNs) were introduced to capture the dynamics of sequences via cycles in the network of nodes [16].

In RNNs, the temporal relations of inputs are addressed by feedback connections for maintaining internal memory states. Recurrent neural networks have proved to be effective in learning time dependent signals for short-term structures [17]. They are capable of storing the previous records in their memory. However, in large time series sequences, the vanishing gradient hampers the learning of these models. This problem occurs when the gradient updates become very small and add no major contribution towards model learning. The LSTM neural networks were introduced to overcome the vanishing gradient problems of RNNs with capability of storing important information containing long sequences [18]. To further improve the performance of RNNs, more advance LSTMs were developed by Schuster and Paliwal [19], named bidirectional LSTM. The LSTMs relate different past records in time series problems; however, in bidirectional LSTMs the enhanced learning mechanism enable them to relate the past as well as future records for better estimation of a variable.

The LSTMs are used in a number of applications including speech recognition, time series predictions, and grammar learning. However, the application of LSTMs in the field of hydrology has not been widely reported in literature but can be used in estimation of hydrologic variables since several climatic variables used in hydrology exhibit time series behavior. Several studies have highlighted the potential of LSTMs in rainfall runoff modeling [20,21] in which the performance of LSTMs was better than physically-based runoff models. Zhang et al. [22] used LSTMs for groundwater estimation and reported that they performed better than the multilayer perceptron model. They used the dropout effect in hidden layers of LSTMs to increase the model learning for better estimation of groundwater. However, based on the literature review, the use of advanced RNNs in ETo modeling is very limited and/or unpublished. Need for this study is based on the fact that out of 57 meteorological weather stations, installed in Prince Edward Island, less than 10 stations provide enough data for ETo estimation for the FAO-56 method. This leaves researchers and the government sectors, responsible to promote sustainable agriculture in the island, to look for other approaches of estimating ETo to guide

farmers about making intelligent decisions for irrigating their crops including potatoes. Potato industry significantly promotes the economy of Prince Edward Island as it contributes about 10.8% to the GDP (gross domestic product) of this province with more than one billion dollars direct and indirect economic benefits, while engaging 12.1% of the total work force of the island [23]. Currently, Prince Edward Island produces approximately 20–25% of total potatoes grown in Canada each year [24]. The potato is a very sensitive crop in terms of both yield and quality under limited water conditions [25] and soil water should not be depleted by more than 30–50% for optimum potato yield [26,27]. The findings of Shock et al. [27] revealed the high risk of reduced potato yields under water shortage conditions. The majority of potato production in Prince Edward Island is rainfed and irregular rainfall patterns may affect potato yield. Furthermore, climate changes add severity to this problem as more hot days, less cold days, and changes in precipitation patterns have been predicted for this island during the next decades [28]. These challenges demand the correct quantification of plant water requirements, calculated from ETo, through the use of robust and accurate methods. The neural networks and analysis used in this study to estimate ETo have not been previously used and/or published for this region making this study novel and innovative for the scientific community and for government sectors involved in agricultural activities.

Therefore, this study explores the performance of conventional LSTMs and bidirectional LSTMs in modeling ETo across Prince Edward Island, Canada with the specific objectives to (i) model ETo with high accuracy as well as with a reduced number of input variables and (ii) assess the need of supplemental irrigation by comparing the rainfall and estimated ETo for sustainable production of potatoes in Prince Edward Island.

## 2. Experiments and Methods

### 2.1. Site Selection

Prince Edward Island is an Atlantic Canadian province, situated in the Gulf of Saint Lawrence and separated from the other Atlantic provinces, namely Nova Scotia and New Brunswick, at Northumberland Strait. The climate of Prince Edward Island is considered as humid, which is strongly influenced by the surrounding seas and their variants throughout the year. The island winter season is long with a relatively shorter summer season. In winter, the island receives storms and blizzards originating from North Atlantic or Gulf of Mexico. Springtime temperatures are cool, when the ice generally melts in late April. Summers are moderately warm as the daytime temperature occasionally reaches as high as 30 °C (86 °F). The island receives the heaviest rainfall spells of the year in late autumn and during early winter.

Four meteorological sites were selected across the island to represent climatic conditions of the whole island (Figure 1). For example, North Cape (47.058056° N 63.998611° W) was selected to represent the west part of the island. Summerside (46.441111° N 63.838056°bW) and Harrington (46.343617° N 63.169736° W) meteorological sites were selected to represent the central parts of the island. Saint Peter (46.450278° N 62.575833° W) meteorological station represented eastern parts of the island.

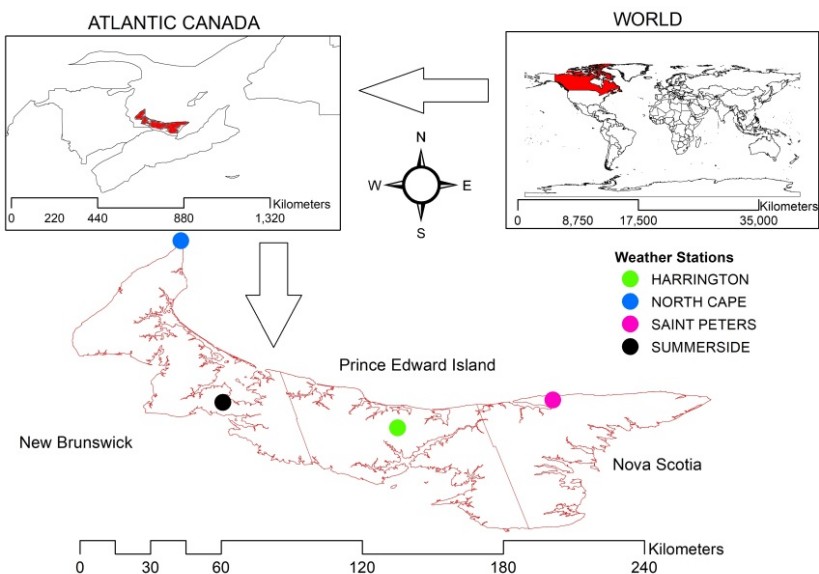

**Figure 1.** Locations of the four selected meteorological stations across Prince Edward Island, Canada.

## 2.2. Data Collection and Variable Selection

Daily climatic data of four selected meteorological sites for period 2011–2017 were retrieved from Environment Canada historical database. Initially, nine different variables, namely heat degree days, hourly mean air temperature, minimum air temperature, maximum air temperature, relative humidity, dew point temperature, wind speed, atmospheric pressure, and daily mean air temperature, were selected. To capture the variation of climatic variables from different parts of the island, a new dataset (PEI dataset) was formed by averaging the variables of all four sites. The 2557 daily records of nine different variables used in this study were collected between the years 2011–2017 for the respective months of the period 2011–2017.

Regression subset analysis was conducted with varied input combinations to select the appropriate inputs for artificial intelligence models. By selecting ETo as a response variable, the nine selected variables were regressed using Minitab (Version 18). Minitab is a statistical package developed by researcher Barbara F. Ryan from Pennsylvania State University. The best performing variables based on the highest coefficient of determination ($R^2$) were selected as input variables for RNNs (Table 1).

**Table 1.** The structures of recurrent neural networks for analysis of this study finalized after several trials.

| Recurrent Neural Networks | Components of Recurrent Neural Networks | Used Components after Trials |
|---|---|---|
| Long short-term memory neural networks | Hidden layers | 1 |
| | Forward layers | 1 |
| | Backward layers | 0 |
| | Neurons | 100 |
| | Learning rate | 0.01 |
| | Optimizer | Adam |
| | Batch size | 128 |
| Bidirectional long short-term memory neural networks | Hidden layers | 2 |
| | Forward layers | 1 |
| | Backward layers | 1 |
| | Neurons | 100 |
| | Learning rate | 0.001 |
| | Optimizer | Adam |
| | Batch size | 256 |

### 2.3. Penman–Montieth FAO-56 Model

Actual ETo data were not available for the selected study sites; therefore, the FAO-56 method was used to estimate ETo for these sites. The estimated ETo from the FAO-56 method was used as targets for LSTMs and bidirectional LSTM neural networks. The FAO-56 model is accepted and has been widely used [3,29,30] in these situations, and is derived from:

$$\text{ETo}\left(\frac{mm}{day}\right) = \frac{\Delta(R_n - G) + \rho_a c_p\left(\frac{es-ea}{ra}\right)}{\Delta + \gamma\left(1 + \frac{r_s}{r_a}\right)}. \tag{1}$$

The Penman–Monteith equation (Equation (1)) was derived from the most acceptable form by Allen et al. [3] also known as the FAO-56 model expressed as Equation (2) below:

$$\text{EETo}\left(\frac{mm}{day}\right) = \frac{0.408\Delta(R_n - G) + \gamma\left(\frac{900}{Tmean+271}\right)U(es - ea)}{\Delta + \gamma(1 + 0.34U)}, \tag{2}$$

where $\Delta$ is the slope of the vapor saturation pressure, $R_n$ is the net radiation, $G$ is the soil heat flux, $\rho_a$ is the mean air density at constant air pressure, $C_p$ is the specific heat of the air, $es - ea$ is the vapor pressure deficit, $\Upsilon$ is the psychrometric constant, $U$ is wind speed at 2 m (m/s), *Tmean* is daily mean temperature, $\Upsilon_s$ is the surface resistance, and $\Upsilon_a$ is the aerodynamic resistance *m*.

### 2.4. Long Short-Term Memory Neural Networks

The RNNs are sequence-based models, equipped with memory blocks to store and relate the previous information in a sequence. However, vanishing gradient hinders the learning in earlier layers of RNNs and this phenomenon is sometimes referred as short-term memory. Input ($X_t$), output ($o_t$), and forget ($f_t$) gates were added in memory blocks of LSTMs to address short-term memory problems. The forget gate has the ability to discard irrelevant information based on relevance; i.e., the input variables after normalization closer to 0 are forgotten and closer to 1 are kept for further use. Forget gate in LSTMs reduces the chances of overfitting by not carrying out all information from the previous steps. Selective information control in LSTM is the key reason to overcome the vanishing gradient problems and make them suitable for nonstationary data modeling. After passing through $f_t$, tanh and sigmoid functions are used to scale the values for further processing. Combine state ($C_t$) is computed as a result of dot product between tanh and sigmoid outputs. The more detailed overview of the LSTM information flow memory block is described in Figure 2.

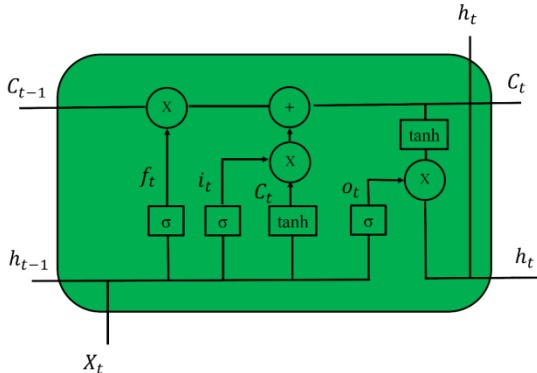

**Figure 2.** The memory block of long short-term memory (LSTM) neural networks.

### 2.5. Bidirectional Long Short-Term Memory Neural Networks

Bidirectional LSTMs have two-way information flow in contrast with traditional LSTM (Figure 3). Bidirectional LSTMs can relate information from previous as well as future time steps, making them

more powerful than traditional LSTMs. The outputs from both directions then aggregate for labels prediction. The memory blocks of bidirectional LSTM work similar to traditional LSTMs (Figure 2).

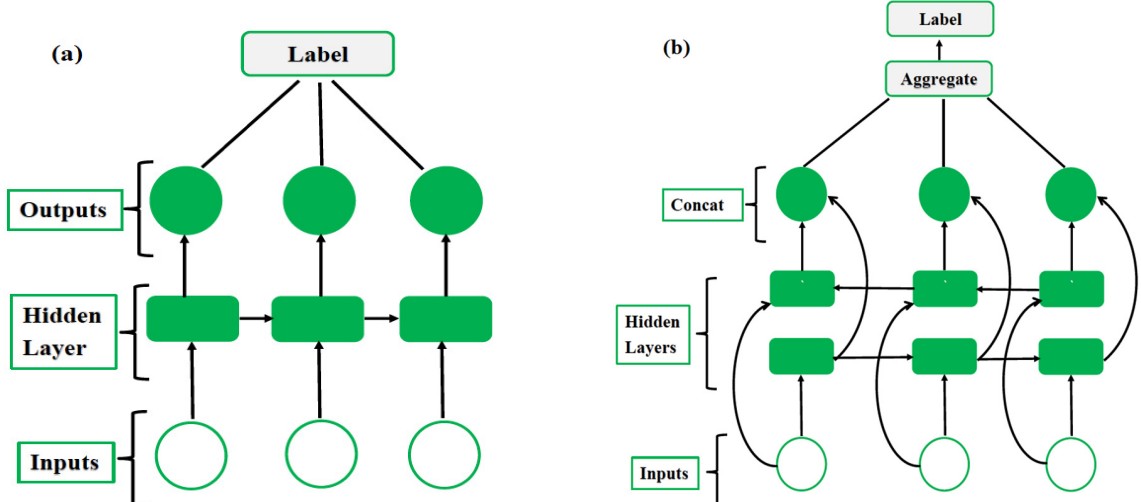

**Figure 3.** The basic structure of (**a**) long short-term memory neural networks (LSTM) and (**b**) bidirectional LSTM.

### 2.6. Hyperparameter Tuning and Reproducibility

Daily meteorological variables were split in training (2011–2015) and testing (2016–2017) sets. The test sets were used to estimate the ETo after successful (convergence) training of the models. Extensive tests were performed to determine the hyperparameters of LSTMs. Several hyperparameters including neurons, learning rates, optimizers, batch sizes, and dropout effects were tested for higher accuracy. The highest performing learning rates, neuron and batch sizes were used in models training and testing. Due to large data points used in this study, several data normalization techniques were tested to reduce the noise effects in data. The max-min normalization performed better with our data in comparison with standard scalar, absolute scalar, power transformer, and normalizers. After training of models, the data were back transformed to the original scale.

The TensorFlow framework was selected because of its wide applications in industrial deployment. TensorFlow is an open source library developed by the Google Brain team in November 2015. TensorFlow (Version 2.0) was used in the study along with Keras. Keras is another open source library developed by a Google engineer, Francois Chollet. The purpose of Keras was to simplify the neural net modeling for user community. The other libraries used in this study included Numpy version 1.18.1 (community project), Matplotlib version 3.1.3 (developed by John D. Hunter), Pandas version 1.0.1 (community project), and Scikitlearn version 0.22.1 (developed by David Cournpeau) used with Python version 3.6 (developed by Guido van Rossum) programming language. All models were trained using Dell Latitude 5580 workstation, with Intel Core I7 7600U CPU, 8GB ram, Nvidia GeForce 930MX, and Ubuntu 16.04 x 64 operating system (developed by Mark Richard Shuttleworth). Furthermore, for reproduceable results, all random seeds were set to avoid randomness. For examples, Python-hash seeds were set to 0, Numpy random seeds were set to 111, Python random seeds were set to 10, and TensorFlow random seeds were set to 89. All the results displayed in this study were retrieved using the above mentioned random seed configuration.

### 2.7. Rainfall Evapotranspiration Comparison

Evapotranspiration is considered to be the prominent parameter in the water balance equation, which is expressed as:

$$P = Q + \text{ETo} + \Delta S, \tag{3}$$

where *P* is precipitation, *Q* is runoff, ETo is evapotranspiration, and Δ*S* is represented as storage in soil. Over the longer periods, changes in water storage for a particular region may be neglected [31] and precipitation is balanced by runoff and ETo only. Furthermore, in agricultural, the land runoff has minor effects on the water balance equation because of higher infiltration rates of soils. Soils of Prince Edward Island are sandy loam and exhibit negligible surface runoff and soil water retention. The finding of a study by Carter [32] suggested that the majority of soils in agricultural production are sandy loam with relatively less percentages of clay in comparison with sands. Therefore, rainfall and ETo were compared on the province scale without considering the effect of runoff and change in soil storage. The potato growing season (June–November) was considered to compare rainfall with ETo, as the agriculture is not possible in winter season [33] in Prince Edward Island because of the historical snow and colder weather conditions. Furthermore, the winter season soil water storage (Δ*S*) should not be neglected as it may influence the water balance even for longer periods because of snow accumulations during winter and snow melt during early summer. Due to the above said reasons, the limitation of the present study is that the rainfall and ETo analysis used in this study is valid only for potato production season (June–November).

## 2.8. Model Evaluation

Loss of the model was evaluated by mean absolute error (MAE), which is the average of all absolute errors between the predictions and labels. MAE is expressed as:

$$\text{MAE} = \frac{1}{N} \sum_{i=1}^{N} |y_i - \hat{y}i|. \tag{4}$$

The root means square error (RMSE) and $R^2$ were used to evaluate the model effectiveness as these two evaluation parameters have been used in various studies to evaluate the neural networks predictive power [22,33,34]. Values of $R^2$ closer to 1 represented the models with higher predictive power. RMSE and $R^2$ were defined as:

$$\text{RMSE} = \sqrt{\frac{\sum_{i=1}^{N} (yi - \hat{y}i)^2}{N}}, \tag{5}$$

$$R^2 = \sqrt{\frac{\sum_{i=1}^{N} (yi - \overline{y})^2 - \sum_{i=1}^{N} (yi - \hat{y}i)^2}{\sum_{1=1}^{N} (yi - \overline{y})^2}}, \tag{6}$$

where $y_i$ is the actual value at the *i*th time, $\hat{y}_i$ is the estimated value at the *i*th time, and *i* ranges from 1 to *N*. $|y_i - \hat{y}_i|$ are the absolute error between actual and predicted values at *i*th time.

## 3. Results and Discussions

### 3.1. Selection of Climatic Variables

The results of subset regression analysis suggested that the maximum air temperature was the highest contributor among selected variables in estimation of ETo. For all sites, $R^2$ was in the range of 0.70–0.74 between ETo and maximum air temperature. Higher $R^2$ indicated the strong predictive power of maximum air temperature in estimation of ETo. A study by Feng et al. [35] explained the significance of temperature data in reference to ETo modeling. They used only temperature data to estimate ETo. These results are in agreement with the findings of Feng et al. [35] and confirmed the relevance of temperature data for modeling ETo. Relative humidity was the second largest contributor in estimation of ETo as it increased the $R^2$ by 0.11%, 0.13, 0.14, 0.10, and 0.12 for Saint Peter, Harrington, North Cape, Summerside, and PEI data sets, respectively (Table 2). By increasing the number of variables from 2 to 5, there were minor increases of 0.02, 0.01, 0.02, 0.02, and 0.02 in $R^2$ for Saint Peter, Harrington,

North Cape, Summerside, and PEI data sets, respectively. Based on subset regression analysis results, only two variables, namely maximum air temperature and relative humidity, were selected in training of recurrent neural networks. One of the objectives of this study was to decrease the number of variables to a possible extent without compromising the overall accuracy. A study by Afzaal et al. [34] concluded that there was no major improvement in deep learning models predictive accuracy by increasing the number of variables from 2 to 4.

### 3.2. Descriptive Statistics of Selected Input Variables

Descriptive statistics of the selected variables are given in Table 2. The maximum air temperature ranged between −17 and 33.5 °C for the period 2011–2017 for all sites. The highest maximum air temperature was recorded to be 33.5 °C for Summerside. The mean of maximum air temperature was in the range of 9.2–10.7 °C with high standard deviation; i.e., 10.2–10.8 °C for all sites. Because of seasonality, the maximum air temperature behaved as bimodal distribution for all sites (Figure 4). Exponential relation of maximum air temperature with ETo is evident in Figure 4.

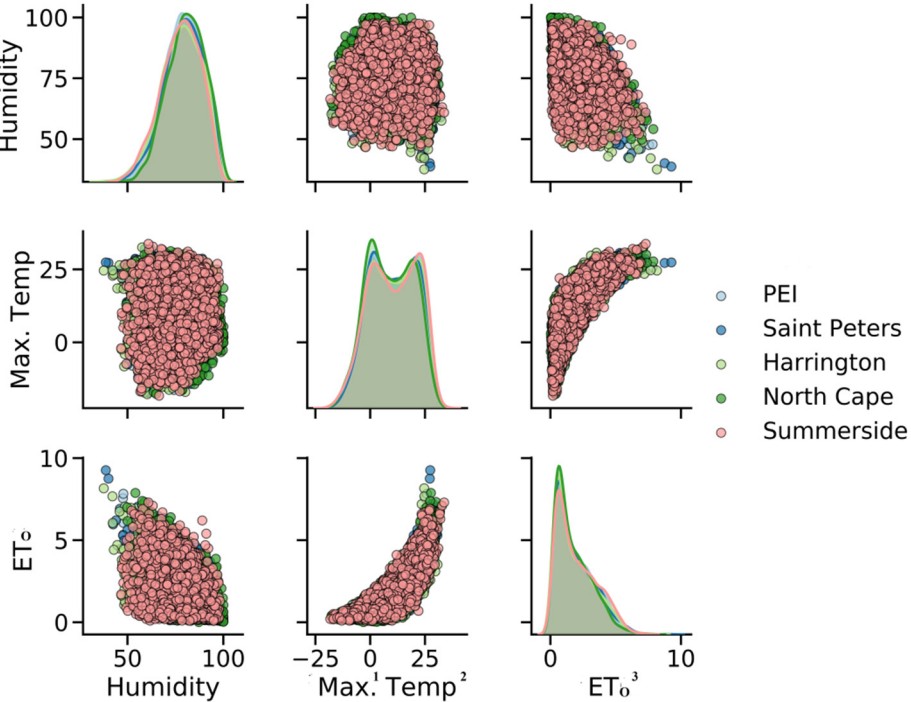

**Figure 4.** Pair plots of selected variables for year 2011–2017. Where [1]Max. Temp is Maximum air temperature; [2]ETo is reference evapotranspiration computed from the FAO-56 method.

**Table 2.** Subset regression analysis regressed versus FAO-56 ETo.

| Site | Variable (Number) | $R^2$ | [1] HDD | [2] Hourly Mean Air Temp (°C) | [3] Min. Air Temp (°C) | [4] Max. Air Temp (°C) | Relative Humidity (%) | [5] Dew Point Temp (°C) | [6] Daily Mean Air Temp (°C) |
|---|---|---|---|---|---|---|---|---|---|
| Saint Peters | 1 | 71.9 | | | | X | | | |
| | 2 | 83.0 | | | | X | X | | |
| | 3 | 83.9 | | | | X | | X | X |
| | 4 | 84.6 | X | | | X | | X | X |
| | 5 | 85.2 | X | X | | X | | X | X |
| Harrington | 1 | 70.7 | | | | X | | | |
| | 2 | 83.9 | | | | X | X | | |
| | 3 | 84.4 | | | | X | | X | X |
| | 4 | 84.8 | X | | | X | | X | X |
| | 5 | 85.2 | X | X | | X | | X | X |
| North Cape | 1 | 72.5 | | | | X | | | |
| | 2 | 87.0 | | | | X | X | | |
| | 3 | 88.0 | | | | X | | X | X |
| | 4 | 88.8 | X | X | | X | X | | |
| | 5 | 89.4 | X | X | | X | | X | X |
| Summerside | 1 | 73.6 | | | | X | | | |
| | 2 | 83.6 | | | | X | X | | |
| | 3 | 84.2 | | | | X | | X | X |
| | 4 | 84.7 | X | | | X | | X | X |
| | 5 | 85.2 | X | | X | X | | X | X |
| PEI | 1 | 74.4 | | | | X | | | |
| | 2 | 86.2 | | | | X | X | | |
| | 3 | 87.2 | | | | X | | X | X |
| | 4 | 87.7 | X | | | X | | X | X |
| | 5 | 88.3 | X | | X | X | | X | X |

[1] Heat degree days; [2] Hourly mean air temperature; [3] Minimum air temperature; [4] Maximum air temperature; [5] Dew point temperature; and [6] Daily mean air temperature.

Relative humidity ranged from 37.6% to 100% for all sites. The average humidity was recorded to be 77.7–80.1% with higher standard deviation of 37.6–49.3% in year 2011 to 2017 for all sites (Table 2). Relative humidity represented with normal distribution and weaker inverse relation between ETo and relative humidity may be visualized in Figure 4.

ETo computed from the FAO-56 method ranged between 0 and 9.3 mm/day for all sites and duration of 2011–2017. The maximum daily $ET_O$ was recorded to be 9.3 mm/day for Saint Peter site. The mean daily ETo ranged from 1.8 to 2.0 mm/day with slightly lower standard deviation of (i.e., 1.4–1.5 mm/day) for all sites. Distribution of daily ETo may be represented with right skewed distribution because of relatively low values in the winter season (Figure 4).

The rainfall varying from 0 to 147.5 mm/day was recorded for all sites in the study period. The highest rainfall of 147.5 mm was recorded for the North Cape site. The average rainfall received by all sites was in the range of 2.4–3.2 mm/day with standard deviation of 6–7.7 mm/day. The Summerside station received relatively less rainfall in comparison with other sites (Table 3).

**Table 3.** Descriptive statistics of input and output variables for year 2011 to 2017.

| Variable | Site | Mean ± SD | Minimum | Maximum |
|---|---|---|---|---|
| Maximum Temperature (°C) | Harrington | 10.5 ± 10.6 | −17.7 | 32.5 |
| | North Cape | 9.2 ± 10.2 | −17.6 | 31.2 |
| | PEI | 10.2 ± 10.5 | −17.2 | 31.8 |
| | Saint Peters | 10.5 ± 10.4 | −17.0 | 32.0 |
| | Summerside | 10.7 ± 10.8 | −18.2 | 33.7 |
| Reference Evapotranspiration (mm/day) | Harrington | 1.9 ± 1.5 | 0.1 | 8.2 |
| | North Cape | 1.8 ± 1.4 | 0.0 | 7.9 |
| | PEI | 1.9 ± 1.4 | 0.1 | 7.8 |
| | Saint Peters | 1.9 ± 1.5 | 0.1 | 9.3 |
| | Summerside | 2.0 ± 1.5 | 0.1 | 7.3 |
| Rainfall (mm/day) | Harrington | 3.1 ± 7.0 | 0.0 | 92.9 |
| | North Cape | 3.1 ± 7.7 | 0.0 | 147.5 |
| | PEI | 3.0 ± 6.0 | 0.0 | 89.7 |
| | Saint Peters | 3.2 ± 7.1 | 0.0 | 85.3 |
| | Summerside | 2.4 ± 6.1 | 0.0 | 103.8 |
| Relative Humidity (%) | Harrington | 78.1 ± 10.5 | 37.6 | 99.4 |
| | North Cape | 80.7 ± 9.6 | 49.3 | 100.0 |
| | PEI | 78.9 ± 9.5 | 47.6 | 98.4 |
| | Saint Peters | 79.1 ± 10.0 | 38.7 | 98.0 |
| | Summerside | 77.7 ± 10.2 | 46.7 | 98.3 |

*3.3. Model Training and Tesing Evaluation*

In training of RNNs, several optimizers were tested including Stochastic gradient descent, Adam, Adagrad, and RMSprop. The performance of Adam remained better in comparison with other optimizers in terms of accuracy and model convergence. The better performance of the Adam optimizer is in agreement the findings of Reddy et al. [36]. No major effect of increasing the number of neurons on model $R^2$ and RMSE was observed in training of RNNs used in this study. Similarly, no major effect of different learning rates on model $R^2$ and RMSE was observed e.g., $10^{-2}$, $10^{-3}$, and $10^{-4}$. Dropout effect was also tested by freezing the 10%, 20%, and 30% random neurons to reduce the overfitting effects in training of RNNs used in this study. However, because of data normalization of the datasets, the RNNs used in the study were successfully converged without over and under fitting with approximately equal training and testing accuracies. Similar results were found in a study by Afzaal et al., [34] as no major effect of dropout was observed with normalized data. Therefore, all the RNNs used in this study were trained without introducing dropout in LSTM layers.

In training of LSTM models for the Saint Peter site, training and testing losses were recorded to be 0.042 and 0.0404, respectively. Approximately equal values of training and testing losses depict

the successful model convergence without overfitting. LSTM models training RMSE was recorded to be 0.497 mm/day and training $R^2$ was recorded to be 0.88. Similarly, the value of RMSE for testing the LSTM model was recorded to be 0.46 mm/day and testing $R^2$ was 0.91. No major differences were observed in training and testing set accuracies when modeled with bidirectional LSTM for the Saint Peters site (Table 4). It is evident that with both models, there were higher testing accuracies in comparison with training accuracies, maybe because in the training stage usually the ANNs try to adjust their weights. Another reason could be because of unequal data points for training and testing phases. The higher numbers of data points in the training stage (give number of data points) might have reduced the accuracy of RNNs during the training phase.

**Table 4.** Training and testing evaluation of recurrent neural networks.

| Site | Model | Training MAE | Testing MAE | Training RMSE | Training $R^2$ | Testing RMSE | Testing $R^2$ |
|---|---|---|---|---|---|---|---|
| St Peters | LSTM | 0.0420 | 0.0404 | 0.50 | 0.88 | 0.46 | 0.91 |
| | [1] B LSTM | 0.0419 | 0.0405 | 0.49 | 0.88 | 0.46 | 0.91 |
| Harrington | LSTM | 0.0523 | 0.0555 | 0.54 | 0.85 | 0.58 | 0.86 |
| | B LSTM | 0.0461 | 0.0450 | 0.48 | 0.86 | 0.46 | 0.91 |
| North Cape | LSTM | 0.0337 | 0.0380 | 0.35 | 0.93 | 0.39 | 0.92 |
| | B LSTM | 0.0340 | 0.0375 | 0.34 | 0.93 | 0.38 | 0.92 |
| Summerside | LSTM | 0.0550 | 0.0490 | 0.53 | 0.87 | 0.45 | 0.91 |
| | B LSTM | 0.0563 | 0.0497 | 0.53 | 0.87 | 0.45 | 0.91 |
| PEI | LSTM | 0.0417 | 0.0438 | 0.40 | 0.91 | 0.42 | 0.92 |
| | B LSTM | 0.0415 | 0.0437 | 0.40 | 0.91 | 0.42 | 0.92 |

[1] B LSTM; bidirectional LSTM.

For the Harrington site, slightly higher losses and lower accuracies were observed in comparison with the Saint Peters site (Table 3). In training of LSTM for the Harrington site, training and testing losses were recorded to be 0.0523 and 0.0555, respectively. Training and testing RMSE were 0.54 and 0.58, respectively, for LSTM models. The respective training and testing $R^2$ were 0.85 and 0.86, respectively. In training of bidirectional LSTM for the Harrington site, higher training and testing accuracies were recorded in comparison with LSTM. The testing accuracy of bidirectional LSTM was 5% higher in comparison with LSTM for the Harrington site. The advance architecture of bidirectional LSTM might help them to attain the higher accuracies in comparison with LSTM. Furthermore, the dataset of the Harrington sites showed slightly higher standard deviation in comparison with other sites selected for this study. The results suggested that the bidirectional LSTM can achieve higher accuracy with scatter data in comparison with LSTM.

North Cape LSTM training and testing losses recorded to be 0.0337 and 0.0380, respectively; slightly lower than all other sites. The LSTM training and testing RMSE was recorded to be 0.35 and 0.39, respectively. The LSTM training and testing $R^2$ was recorded to be 0.93 and 0.92, respectively. No major differences were observed in losses and accuracies for the North Cape site when modeled with bidirectional LSTM. The two directional learning may achieve the higher accuracies in the time series forecasting problem. However, this study estimates the time steps of time series by inputting climatic variables only as there are no forecasting involved.

Summerside LSTM training and testing losses were recorded to be 0.0550 and 0.0490, respectively. The LSTM training and testing RMSE was recorded to be 0.53 and 0.45, respectively. The LSTM training and testing $R^2$ was recorded to be 0.87 and 0.91, respectively. No major differences were observed in losses and accuracies for the Summerside site when modeled with bidirectional LSTM. For both RNN, higher testing accuracies were observed in comparison with training accuracies for Summerside.

For the PEI data set, slightly lower LSTM training and testing losses were observed (Table 3). The LSTM training and testing $R^2$ were recorded to be 0.91 and 0.92 respectively. There were no major differences in losses and accuracies for PEI data set when modeled with bidirectional LSTM (Figure 5).

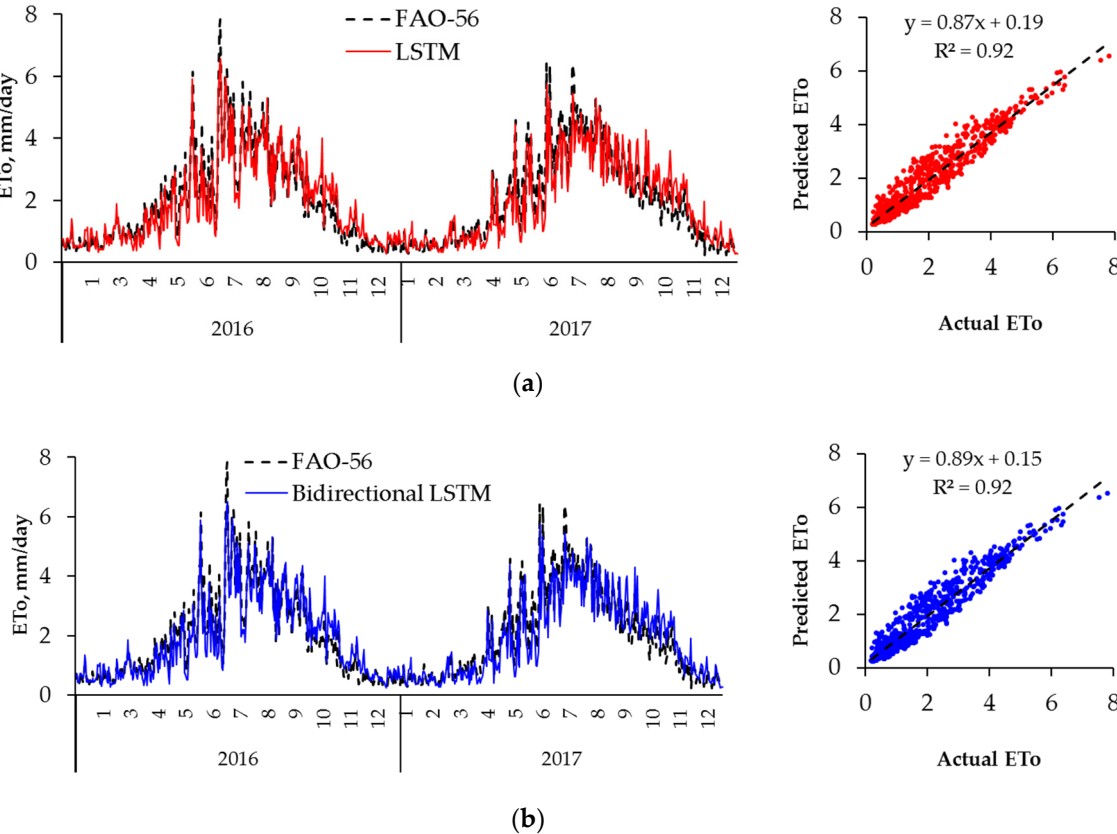

**Figure 5.** (**a**) Comparison of ETo predicted with LSTM and FAO-56, (**b**) with bidirectional LSTM and FAO-56 using combined data set of Prince Edward Island for the test period 2016–2017.

Overall, no major effect was observed in the accuracy of LSTM and bidirectional LSTM for all sites. However, for Harrington and PEI data sets, the accuracy of bidirectional LSTM was slightly better. In another similar study of turbulent flow modeling, Mohan and Gaitonde [37] also found better performance of LSTM in comparison with bidirectional LSTM. However, similar performance of LSTM and bidirectional LSTM was observed except for two sites (Harrington and PEI combined data) in which performance of bidirectional LSTM was better than LSTM.

Mehdizadeh et al. [38] tested several methods, namely gene expression programming, support vector machine, multivariate adaptive regression splines for ETo modeling. With selected two variables combinations (relative humidity and temperature), they had slightly higher RMSE and lower $R^2$ in comparison with the RNN models used in this study. In another study, Patil et al. [13] tested the extreme learning approach in ETo modeling. Their reported RMSE, by using three variables (minimum temperature, maximum temperature, and solar radiation), were slightly higher than the RMSE of RNNs used in this study. The relatively new methods for ETo modeling used in this study performed better than the previous techniques. However, a detailed comparative analysis is required for further evaluation of these models.

### 3.4. Rainfall and Reference Evapotranspiration Comparison

One of the objectives of this study was to highlight the gap between ETo and rainfall in order to strategize the need for supplemental irrigation and sustainable agriculture. A comparison between the cumulative values of rainfall and ETo for the period 2011–2017 is displayed in Figure 6 to gauge the gap between the two variables. The results showed a high variability of rainfall in different months (of the growing season) during all years of the study period than variability in ETo. For the month of June, the rainfall ranged between 64.85–107.025 mm while FAO-56 ETo ranged between 83.82–112.47 mm with an average difference of 13.52 mm. The ETo clearly surpassed the rainfall values in the month of

June for periods 2011–2014 and 2016–2017. The highest gap between rainfall and ETo was observed in the month of July for year 2012–2017. In July, the recorded difference between ETo and rainfall was computed to be 3.66, −92.2, −56.1, −85.0, −73.4, −68.5, −98.7 mm for years 2011 through 2017, respectively. The negative values clearly show the higher ETo than the respective rainfall. In the month of August, the ETo surpassed rainfall in the years 2012, 2013, 2015 2016, and 2017 by 46.1, 30.0, 4.39, 15.8, and 39.6 mm, respectively. In the months of September, October, and November, rainfall clearly surpassed the ETo. These are the months of crop harvest when crops do not need rainfall and have no ETo phenomenon.

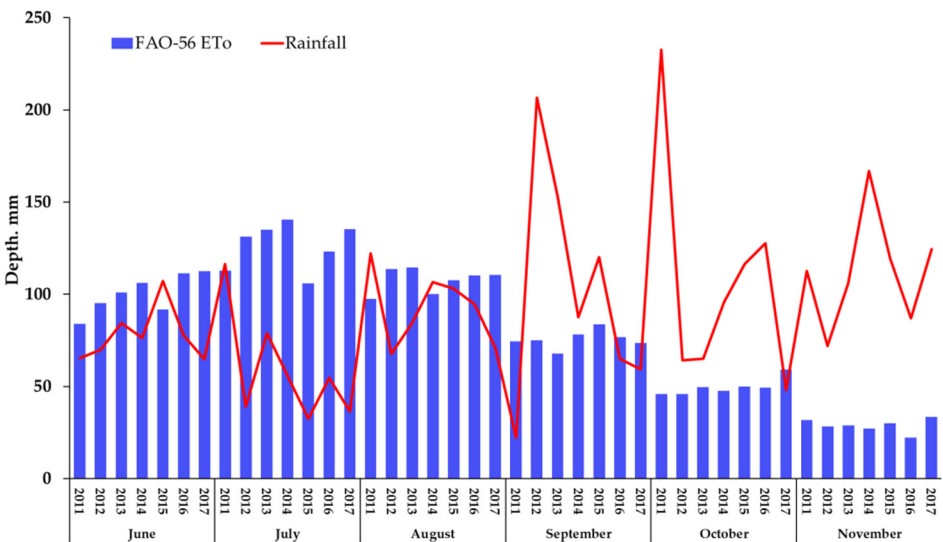

**Figure 6.** A comparison of rainfall and FAO-56 ETo for the study period 2011–2017.

The result of this analysis suggested that rainfall is highly variable in the same months of different years unlike ETo, which seems to be more consistent in different years. However, in order to fulfill the crop water requirements, careful monitoring of rainfall and ETo is required in the months of June, July, and August in Prince Edward Island for potential crop yield.

Suitability of the modeling approach adopted in this study was further evaluated by comparing the FAO-56 cumulative ETo with the values simulated using LSTM and bidirectional LSTM in Figure 7, where cumulative rainfall has also been plotted for the test periods 2016 and 2017. A close agreement was found between ETo determined with FAO-56 and the two RNNs. Despite overestimation during the period July–October, yearly cumulative values of ETo determined with all the three methods were also in good agreement with cumulative rainfall during 2016 (Figure 7a).

During the year 2017, higher cumulative gaps were observed between ETo and rainfall in comparison with 2016 (Figure 7b). The lower rainfall during the growing season (June–November) of 2017 (403.74 mm) than during 2016 (506.69 mm) were responsible for these gaps. There was no difference in the trends of RNNs and FAO-56 ETo for the two test years. The cumulative gaps between values of FAO-56 ETo and the values of ETo determined with LSTM and bidirectional LSTM depict the predictive errors of RNNs models. The error may be further reduced by adding more input variables in the RNNs. Furthermore, underestimations for drier months and overestimation for colder months may be removed by adding RMSE correction factors in order to obtain more accurate predictions. The results support the applicability of LSTM and bidirectional LSTM for sustainable water management with accurate estimation of ETo. In order to replenish the crop water requirements, supplemental irrigation might be the option for certain months of the growing season when ETo surpassed rainfall. The estimated ETo will be useful in irrigation scheduling for sustainable cultivation of potato on the island.

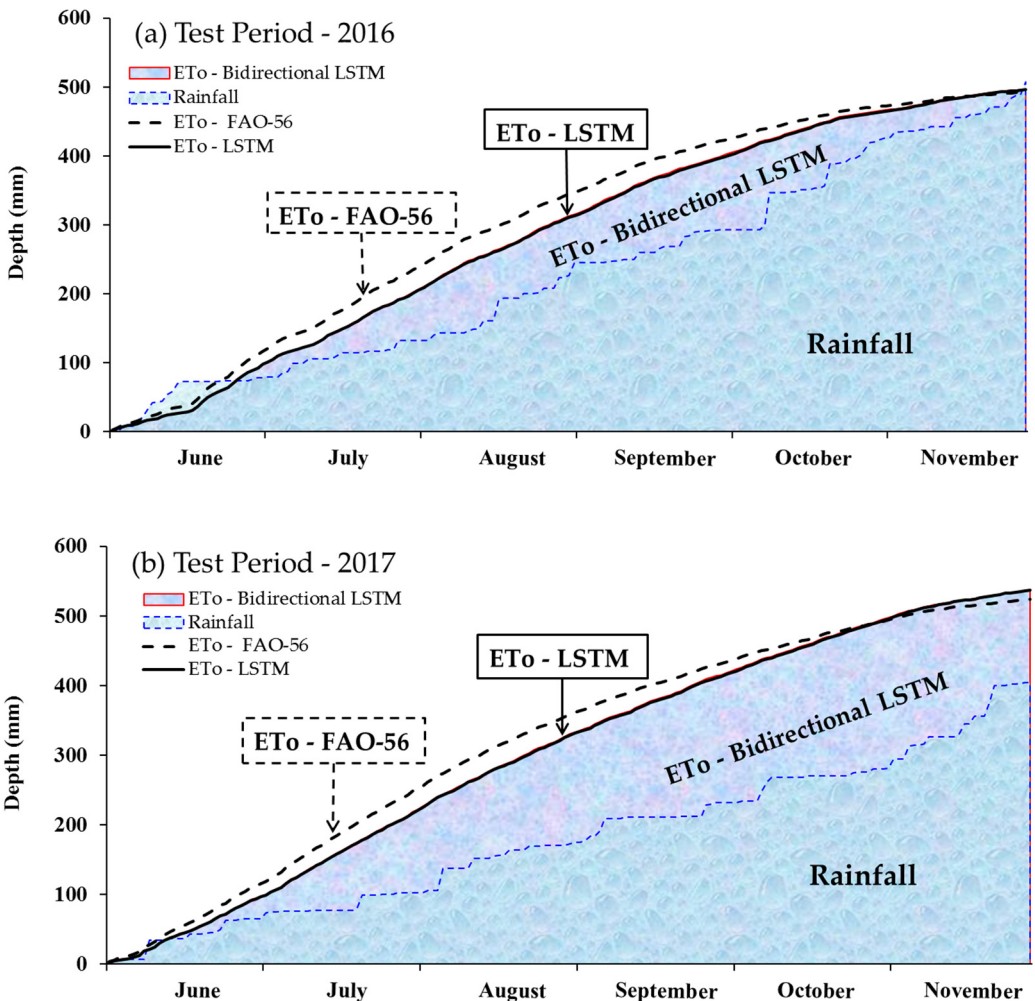

**Figure 7.** From bottom to top, comparison of rainfall (shown as water bubbles area chart separated from ETo-bidirectional LSTM by broken blue lines), ETo-bidirectional LSTM (shown as red vapors separated from $ET_O$-LSTM by solid red line), ETo-LSTM (shown as solid black line), and ETo-FAO-56 (shown as broken black line) for (**a**) year 2016 and (**b**) year 2017. The curve for ETo-BLSTM is hidden under the curve of ETo-LSTM because of negligible difference between the estimates of the two models. This is why ETo-BLSTM has been drawn as an area chart of red vapors above rainfall chart and ETo-LSTM is drawn as a line chart.

## 4. Conclusions

ETo was estimated using FAO-56, LSTM, and bidirectional LSTM at four sites of Prince Edward Island namely Saint Peters, Harrington, Summerside, and North Cape for the study period 2011–2017. Performance of the LSTMs was evaluated using the meteorological data that were split into two sets, namely training set (2011–2015) and testing set (2016–2017). Based on subset regression analysis using nine different climatic variables, the maximum air temperature and relative humidity were selected as inputs for recurrent neural networks. By using tuned hyperparameters, the LSTM and bidirectional LSTM were able to estimate ETo with considerable accuracies determined with ETo calculated using the method of FAO-56. There were no major differences in the accuracy of LSTM and bidirectional LSTM. However, for the Harrington site, bidirectional LSTM performed better in comparison with LSTM for the testing set (2016–2017). The advanced architecture of bidirectional LSTM might be helpful in attaining the higher accuracies in comparison with LSTM. Furthermore, the dataset of Harrington sites showed slightly higher standard deviation in comparison with other sites selected for this study. The results suggested that the bidirectional LSTM can achieve higher accuracy with scatter data in

comparison with LSTM. Another objective of this study was to quantify the difference ETo and rainfall. The analysis showed that in months of June, July, and August, the ETo surpassed rainfall emphasizing the need for viable options such as supplemental irrigation to replenish the crop water requirements in drier months for agriculture sustainability of potato production in Prince Edward Island.

**Author Contributions:** Conceptualization, H.A., A.A.F., and F.A.; methodology, H.A. and F.A.; software, H.A. and F.A.; validation, A.A.F.; formal analysis, A.A.F., B.A., and T.E.; investigation, H.A. and F.A.; resources, A.A.F.; data curation, H.A., B.A., and T.E; writing—original draft preparation, H.A. and A.A.F.; writing—review and editing; B.A., and T.E.; supervision, A.A.F.; project administration, A.A.F. and F.A.; funding acquisition, A.A.F. All authors have read and agreed to the published version of the manuscript.

**Funding:** This research was supported by the Natural Science and Engineering Research Council of Canada, the Prince Edward Island Potato Board, the Canadian Horticultural Council, Potato Board New Brunswick, the New Brunswick Department of Agriculture, Aquaculture and Fisheries (CAP program), and Agriculture and Agri-Food Canada.

**Acknowledgments:** The authors thank and the Precision Agriculture Team at the University of Prince Edward Island for their cooperation and assistance during the experiment.

**Conflicts of Interest:** The authors declare no conflict of interest.

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
