# Peer review of "Computation of Evapotranspiration with Artificial Intelligence for Precision Water Resource Management"

_applsci, doi:10.3390/app10051621_

Round 1

Reviewer 1 Report

The introduction may be overly long providing a great deal of background information that may not be relevant to the entire project. I have a few concerns in how the model was constructed.

The soil type for Prince Edward Island are indicated to be sandy loam with negligible surface runoff and soil water retention. My question is does any of the cropping practices such as minimum tillage plant debris remaining after harvest impact soil water retention during the winter (snow) months? Are all of the soils sandy loam without any water retention during non-production years?

A major climatic variable not included in the modeling was vapor pressure deficit. The two climatic variables in the R2 modeling was temperature and relative humidity for predicting ET. Relative humidity is determined by the relationship between the wet bulb temperature and ambient temperature. Putting relative humidity into the model with maximum air temperature and dew point temperature will of course be a large contributor to the overall model. I would like to see what the model looks like without relative humidity and include only temperature and dew point or vapor pressure deficit. That is more biologically significant.

Author Response

Comment

The introduction may be overly long providing a great deal of background information that may not be relevant to the entire project.

Response

The Introduction section has been restructured for better flow of information and better understanding for the reader. Lines 92-110 were restructures in revised manuscript. 

Comment

I have a few concerns in how the model was constructed. The soil type for Prince Edward Island are indicated to be sandy loam with negligible surface runoff and soil water retention. My question is does any of the cropping practices such as minimum tillage plant debris remaining after harvest impact soil water retention during the winter (snow) months? Are all of the soils sandy loam without any water retention during non-production years?

Response

The reference evapotranspiration (ETo) models were designed on Penman-Montieth (mathematical model) equation with recurrent neural networks. These models do not assume soil water balance equation as the models used relative humidity and maximum air temperatures variables for ETo estimations.

The soil water balance equation was used to compare estimated ETo with rainfall for agriculture sustainability in the island. The surface runoff and water retention conditions were assumed for agriculture production season (June-November) only. In non-production years due to snow accumulation the soil water storage parameters cannot be neglected. The authors agree with reviewer for this shortcoming/limitation in this study as the soil water retention/storage may only be neglected in agriculture production seasons for areas with sandy soil structure.  This text is added in revised manuscript in Lines 222-223 and 227-231.

Comment

A major climatic variable not included in the modeling was vapor pressure deficit. The two climatic variables in the R2 modeling was temperature and relative humidity for predicting ET. Relative humidity is determined by the relationship between the wet bulb temperature and ambient temperature. Putting relative humidity into the model with maximum air temperature and dew point temperature will of course be a large contributor to the overall model. I would like to see what the model looks like without relative humidity and include only temperature and dew point or vapor pressure deficit. That is more biologically significant.

Response

The estimation of vapor pressure involves the use of four variables such as maximum temperature, minimum temperature, minimum relative humidity and maximum relative humidity. The objective of this study was to estimate the ETo with minimal numbers of variables. However, based on reviewer recommendations we tested the suggested variables and got the following results:

Effect of Dew point temperature and Maximum temperature on ETo estimation

Site

Model

Variable

Train R2

Train RMSE

Test R2

Test RMSE

Saint peters

LSTM

Dew point temp

0.535

0.978

0.479

1.101

Saint peters

BLSTM

Dew point temp

0.557

0.957

0.511

1.067

Saint peters

LSTM

Max temp

0.79

0.657

0.765

0.74

Saint peters

BLSTM

Max temp

0.802

0.639

0.786

0.707

Saint peters

LSTM

Dew point and max temp

0.842

0.571

0.843

0.605

Saint peters

BLSTM

Dew point and max temp

0.855

0.547

0.863

0.566

PEI

LSTM

Dew point temp

0.577

0.898

0.521

1.01

PEI

BLSTM

Dew point temp

0.597

0.876

0.558

0.974

PEI

LSTM

Max temp

0.81

0.601

0.776

0.693

PEI

BLSTM

Max temp

0.825

0.578

0.801

0.654

PEI

LSTM

Dew point and max temp

0.883

0.471

0.87

0.527

PEI

BLSTM

Dew point and max temp

0.895

0.448

0.887

0.492

Similar results were recorded for other sites as well. The results revealed the highest contribution of maximum temperature in ETo estimation, e.g. 0.79-0.825 for Saint Peters and PEI sites. The effect of Dew Point temperature on ET estimation was relatively lower in comparison with maximum air temperature. The combined effect of Dew point temperature and maximum air temperature ranged from 0.842-0.895, which was relatively lower than the combined effects of Maximum Temperature and Relative Humidity.

The subset regression analysis was used to assess the individual as well as combination of input variables contribution toward ETo variability. The highest contributed input/inputs combination/combination is/are presented in Table 1. The first row for each site represents the highest contributed one variable towards ETo. The second row represents the best combination of two variable followed by 3, 4 and 5 best variable combinations. For modelling we only chose two variables including maximum temperature and relative humidity as these variables are easily available across meteorological stations. The detailed discussion has been added in section “3.1 Selection of climatic variables” of this manuscript (L242-258).

The estimation of ETo with minimal variables also makes this study easy for practical applications (for irrigation and water resource management). Moreover, the parameters used to calculate ETo were directly obtained from weather stations, making it easy to use for agricultural community.

Reviewer 2 Report

Afzaal and colleagues used recurrent neural net (RNN) to estimate evapotranspiration (ET) using meteorological data collected from ground stations located in a Canadian island. Overall, the topic is relevant given the importance of sustainable agriculture under the changing climate. The authors included enough data and developed prediction models with relatively new neural net-based approaches. The results have some values for the community and broad readers of the journal. However, at this stage, I can only recommend a major revision and for my detailed comment please see below.

In general, the introduction is well-structured. the last part. not in a good position. Based on the content of this para. it could be repositioned as the second para. of the intro, or present it just before the objectives.

ET and ET0, what are the differences? Define and use them consistently throughout.

Include the climate of the study site. Also, clarify the exact months that data collected and a number of samples per year, months, days, hours.

Minitab? provide developer information as well if it is software, the same applies to other software tools and programming languages in the manuscript.

Clarify how did you choose the input variables, it seems you combined different variables for selection.

In figure 3b, your hidden layer not connected! This is not an original BLSTM architecture, revise!

Delete ‘long’ inline 174.

In a table form, maybe, provide final LSTM network architecture e.g, the number of hidden layers, units, backward/forward layers, etc. and hyperparameters and optimization approach used for final models.

Line 183, specify other data normalization functions.

What were the input meteorological data for LSTM models? Were they just max T and RH? Or also rainfall? Clarify!

Also include RMSE% in the manuscript.

Line 327, growing season refers to June – November?

Figure 7 is important but very confusing. The description is related to this figure also inaccurate. Where is the curve for ET0-BLSTM? In figure 7a, I see the overestimation in June not in July and October as in the description. Most of the disagreement between the ET0 and LSTM estimation is between July -October. How do you explain that? Again, a very confusing figure!

In the discussion,  include similar studies that used different approaches and compare their accuracies with yours to highlight the advantages of your approaches.

Also, discuss the limitation of your research.

Line 374, change to advanced!

Author Response

Reviewer 2

Comment

Afzaal and colleagues used recurrent neural net (RNN) to estimate evapotranspiration (ET) using meteorological data collected from ground stations located in a Canadian island. Overall, the topic is relevant given the importance of sustainable agriculture under the changing climate. The authors included enough data and developed prediction models with relatively new neural net-based approaches. The results have some values for the community and broad readers of the journal. However, at this stage, I can only recommend a major revision and for my detailed comment please see below.

In general, the introduction is well-structured. the last part. not in a good position. Based on the content of this para. it could be repositioned as the second para. of the intro or present it just before the objectives.

Response

As per suggestion, the last paragraph of introduction section has been shifted to before objectives (Lines 92-111). The overall flow of introduction is much improved now with this shift.

Comment

ET and ET0, what are the differences? Define and use them consistently throughout.

Response

Thank you for pointing out this inconsistency. To avoid any confusion, we have used the term reference evapotranspiration (ETo) throughout the manuscript by defining it in line 12 of the revised manuscript.  

Comment

Include the climate of the study site. Also, clarify the exact months that data collected and a number of samples per year, months, days, hours.

Response

Lines 121-127: Text about climate of the study site has been added in revised manuscript to describe the climate of island. Lines 143-145 have been added in the revised manuscripts to show the daily records of variables used in this study.

Comment

Minitab? provide developer information as well if it is software, the same applies to other software tools and programming languages in the manuscript.

Response

The Minitab developer information has been added in lines 148-149. Furthermore, all software version and developer information are added in lines 201-210 in revised manuscript.

Comment

Clarify how did you choose the input variables; it seems you combined different variables for selection.

The subset regression analysis was used to assess the effects of the individual as well as combination of input variables on calculations of ETo variability. The highest contributed input/inputs combination/combination is/are presented in newly added Table 1 (Line 197). The first row for each site represents the highest contributed one variable towards ETo. The second row represents the best combination of two variable followed by 3, 4 and 5 best variable combinations.

For modelling, we only chose two variables including maximum temperature and relative humidity as these variables are easily available across meteorological stations. The detailed discussion has been added in the respective section “3.1 Selection of climatic variables” of this manuscript (L242-258).

Comment

In figure 3b, your hidden layer not connected! This is not an original BLSTM architecture, revise!

Response

The figure 3b has been revised. Now all the hidden layers are connected with inputs and outputs. This suggestion has improved the presentation of bidirectional LSTM.

Comment

Delete ‘long’ inline 174.

Response

Deleted. Currently in line 186.

Comment

In a table form, maybe, provide final LSTM network architecture e.g., the number of hidden layers, units, backward/forward layers, etc. and hyperparameters and optimization approach used for final models.

Response

Table 1 (L197) has been added in the revised manuscript depicting the final structure of RNN. The authors appreciate the reviewer for this useful suggestion.

Comment

Line 183 specify other data normalization functions.

Response

Line 194-196: The max-min normalization performed better with our data in comparison with standard scalar, absolute scalar, power transformer and normalizers.

Comments

What were the input meteorological data for LSTM models? Were they just max T and RH? Or also rainfall? Clarify!

Response

We have used the maximum temperature and relative humidity as input variables for LSTM models. We did compare the modelled ETo with rainfall in agriculture production season (June-November) only.

Comment

Also include RMSE% in the manuscript.

Response

The normalized RMSE (NRMSE) indicates predictive power of a model in terms of percentage. We computed the Normalized RMSE by following formula:

However, because of highly skewed distribution of ETo (non-normal data), the NRMSE may mis-interpret the predictive power of the models. There are certain methods used with non-normal data; however, these methods possess more errors, which may not be useful to indicate the model prediction ability. The Figure 4 of the manuscript presents distribution of ETo for different sites as shown below.

The highly skewed distribution represents the lower values of ETo in winter months. Because in Prince Edward Island, the winters are of longer duration; this nature of seasoning shifts the distribution towards left-side making the data distribution non-normal. The regions with equal lengths of winter and summer will have normal distribution of ETo. Since the calculated RMSE% for non-normal data shows low model predictability; therefore, it is not good idea to include RMSE% in the manuscript. Instead, RMSE values for training and testing evaluations have been presented. 

Comment

Line 327, growing season refers to June – November?

Response

Potato is major crop in island and potato growing season (June-November) was considered to compare rainfall with ETO, as the agriculture is not possible in winter season in Prince Edward Island because of the historical snow and colder weather conditions. As this study specifically was designed for agriculture sustainability in the island; therefore, ETo and rainfall data have been compared for the potato growing seasons in Prince Edward Island (June-November).

Comment

Figure 7 is important but very confusing. The description is related to this figure also inaccurate. Where is the curve for ET0-BLSTM? In figure 7a, I see the overestimation in June not in July and October as in the description. Most of the disagreement between the ET0 and LSTM estimation is between July -October. How do you explain that? Again, a very confusing figure!

Response

Figure 7 has been improved by adding legend as well as labels for the curves and area charts. The caption of the figure has been rephrased for clarity as, “From bottom to top, comparison of rainfall (shown as water bubbles area chart separated from ETO - Bidirectional LSTM by broken blue lines), ETO - Bidirectional LSTM (shown as red vapors separated from ETO - LSTM by solid red line), ETO – LSTM (shown as solid black line), and ETO - FAO-56 (shown as broken black line) for (a) year 2016 and (b) year 2017. The curve for ETo-BLSTM is hidden under the curve of ETo-LSTM because of negligible difference between the estimates of the two models. This is why ETo-BLSTM has been drawn as an area chart of red vapors above rainfall chart and ETo-LSTM is drawn as a line chart.”

Comment

In the discussion, include similar studies that used different approaches and compare their accuracies with yours to highlight the advantages of your approaches.

Response

Lines 346-354: Additional text has been added to include similar studies (Mehdizadeh et al. [38] and Patil et al. [13]) to compare their results from different approaches with results of this study.

Comment

Also, discuss the limitation of your research.

Response

New text has been added in in revised manuscript in Lines 222-223 and 227-231. The finding of a study by Carter (1987) suggested that the majority of soil in agricultural production are sandy loam with relatively less percentages of clay in comparison with sands.

Furthermore, the winter season soil water storage () should not be neglected as it may influence the water balance even for longer periods because of snow accumulations during winter and snow melt during early summer. Due to above said reasons, the limitations of the present study is that the rainfall and ETo analysis used in this study is valid only for potato production season (June -November).

Comment

Line 374 change to advanced!

Response

Line 414. Changed as suggested

Round 2

Reviewer 2 Report

I appreciate the authors' effort to improve their work. Based on their response to my comments, I believe the manuscript is ready to be published.